# Scalable and Enhanced Hallucination Detection in LLMs using Semantic Clustering

## Abstract

Large language models (LLMs) are increasingly being adopted across various domains, driven by their ability to generate general-purpose and domain-specific text. However, LLMs can also produce responses that seem plausible but are factually incorrect—a phenomenon commonly referred to as "hallucination." This issue limits the potential and trustworthiness of LLMs, especially in critical fields such as medicine and law. Among the strategies proposed to address this problem uncertainty-based methods stand out due to their ease of implementation, independence from external data sources, and compatibility with standard LLMs. In this paper, we present an optimized semantic clustering framework for automated hallucination detection in LLMs, using sentence embeddings and hierarchical clustering. Our proposed method enhances both scalability and performance compared to existing approaches across different LLM models. This results in more homogeneous clusters, improved entropy scores, and a more accurate reflection of detected hallucinations. Our approach significantly boosts accuracy on widely used open and closed-book question-answering datasets such as TriviaQA, NQ, SQuAD, and BioASQ, achieving AUROC score improvements of up to 9.3% over the current state-of-the-art (SOTA) semantic entropy method. Further ablation studies highlight the effectiveness of different components of our approach.

## 1 Introduction

Large language models are witnessing rapid integration across a variety of NLP tasks (Bommarito et al., 2023; Driess et al., 2023; Bang et al., 2023; Zhong et al., 2023; Achiam et al., 2023; Spataro, 2023). However, even widely adopted systems, such as ChatGPT (OpenAI, 2023) and Gemini (TeamGemini et al., 2023) can sometimes generate content that is illogical or inconsistent with the given context—commonly referred to as "hallucination" (Ji et al., 2023). As a result, hallucination detection, which involves the identification of inaccurate information generated by LLMs, has become a topic of high interest in the literature.

For hallucination detection, the focus is shifted towards capturing the semantic properties of the text, minimizing reliance on lexical and syntactical features, as our primary goal is to assess the accuracy of the generated information, regardless of its phrasing. When sampling multiple responses, if an LLM produces semantically inconsistent information in response to the same question, it indicates uncertainty from the model, which can be a sign of hallucination. Leveraging the concept of semantic similarity and uncertainty across meaning distributions to detect hallucinations, (Kuhn et al., 2023) introduced "Semantic Entropy," an unsupervised method that identifies hallucinations by clustering generated responses based on semantic equivalence, followed by calculating the overall semantic entropy from the uncertainty within each cluster. This method has been proven highly effective. However, its main limitation lies in the clustering approach, which relies on Natural Language Inference (NLI) to determine semantic equivalence, as NLI struggles to capture the full range of semantic properties in text (Arakelyan et al., 2024). In addition, NLI models are built using large-scale transformer-based architectures, causing them to be computationally intensive during inference (Percha et al., 2021).

To address these limitations, we propose an optimized semantic clustering approach based on semantic similarity to calculate entropy over meanings. Our approach utilizes sentence embedding to capture nuanced semantic properties in a high-dimensional context, followed by hierarchical clus-

tering. In doing so, we prioritize the token semantics and efficiently cluster the responses from language models (LMs). Improvement in the homogeneity of clusters in turn improves the entropy estimates, resulting in enhanced hallucination detection. The primary contributions of this work are as follows:

- We introduce a versatile black-box framework for automated hallucination detection across diverse LLMs, requiring no access to internal model states or external knowledge, and applicable to any *off-the-shelf* LM.
- Scalability experiments demonstrate our framework's superior efficiency, achieving a 60-fold speedup over SOTA hallucination detection approaches on large-scale settings (e.g., 200 generations).
- Our approach significantly enhances hallucination detection across a diverse set of well-established open and closed-book Question Answering (QA) datasets, including TriviaQA, NQ, SQuAD, and BioASQ. Notably, it achieves up to a 9.3% increase in AUROC on the NQ dataset using Llama-2-7b-chat.
- Comprehensive ablation studies highlight the critical components driving the optimal performance of our method.

This paper is organized as follows: Section 2 presents an overview of the related works, highlighting the importance of semantics in NLG. Section 3 explains the methodology, introducing notation, outlining the problem statement, and describing the technical and theoretical components of our approach. Section 4 covers the experimental setup, including the datasets and models used, while Section 5 provides an analysis of the results and ablation studies. Finally, Section 6 summarizes our findings and suggests potential directions for future research.

## 2 Related Work

Proliferation of LMs in real-world scenarios, e.g., medical and legal domain, is significantly limited due to their ability to fabricate seemingly plausible but unsubstantiated content (Pal et al., 2023; Dahl et al., 2024). Consequently, researchers have addressed this problem from different perspectives, and the majority of approaches can be broadly categorized as black-box, white-box, or gray-box methods.

Black-box methods depend on the output text generated by LMs. For instance, Manakul et al. (2023) hypothesized that if an LM has adequate knowledge of a concept, sampled responses to queries will likely be more consistent and agreeable, whereas significant contradictions/divergence amongst responses indicate hallucination. White-box methods explicitly use the internal states of the models, e.g., hidden layer activations, to detect and mitigate hallucinatory responses (Burns et al., 2022; Li et al., 2024; Azaria and Mitchell, 2023). Gray-box approaches act as a middle ground and remain oblivious to the internal state of the model while using token probabilities to derive additional metrics, such as confidence scores or predictive uncertainty for detecting hallucinations (Xiong et al., 2023; Xiao and Wang, 2021; Yuan et al., 2021). Another category of approaches aims to detect hallucination by comparing the LLM output with external knowledge sources to verify the truthfulness of the claim (Thorne et al., 2018; Guo et al., 2022). However, these methods introduce dependency on an external source, while being limited by the scope and accuracy of facts in the knowledge repositories. Furthermore, hallucinations also involve subtle reasoning errors that surpass simple fact verification (Kryscinski et al., 2019; Maynez et al., 2020).

Though white-box methods have outperformed black/gray-box tools (Zhu et al., 2024), the improvement is marginal (Xiong et al., 2023), and there is exclusive dependence on the internal state of the model. These are not readily available to users with restricted API usage, and practically challenging to obtain with proprietary LM systems. In contrast, black/gray-box methods offer a viable alternative due to their implementation simplicity, compatibility with *off-the-shelf* LMs, and independence from model-intrinsic parameters and extrinsic knowledge bases. However, these methods depend on the output text or token probabilities, while ignoring the text semantics. Lately, Kuhn et al. (2023) showed that the accuracy of gray-box based hallucination detection can be improved by considering the underlying text semantics. Particularly, 'semantic entropy' was introduced to measure model uncertainty by adjusting for the meaning of a text. This idea of semantic entropy has proven to be

very effective in hallucination detection, and we introduce a brief background on the importance of semantics in Natural Language Generation (NLG).

**Semantics in NLG** The complexities associated with natural language mean that identical subjects can be expressed in many different ways. It is essential to first distinguish between semantics, syntax, and lexical content. As defined in the literature, syntax involves the grammatical properties of the text, lexical content involves the words used within the text, while semantics involves the overall intended meaning (Lyons, 1995). In NLG, particularly within the context of hallucination detection, we prioritize the semantic properties of the text, to determine the likelihood of potential inaccuracies and/or inconsistencies. When presented with a question, a model is able to address this question in more ways than one, while still maintaining a level of reliability and accuracy. As a result, it is important for us to effectively capture and understand semantic properties of text as an indication of generation reliability.

**Significance of Semantics in Estimating Model Uncertainty** Kuhn et al. (2023) proposed an interesting viewpoint for estimating the uncertainty in LM models, specifically where different sentences can mean the same thing and 'syntactic difference may not imply different semantics'. A sentence can be phrased differently and have different form or syntax, without changing its underlying semantics - a phenomenon referred to as 'semantic equivalence'. For example, the two sentences: 'rhinovirus are the predominant cause of common cold' and 'common cold is caused by rhinovirus' have the same meaning. However, at the level of token likelihood, if the model is uncertain about which sentence to generate, this uncertainty is semantically insignificant. Consequently, Kuhn et al. (2023) used semantic equivalence to induce a probability distribution over the meaning of tokens (instead of lexical structure) to capture the semantic uncertainty. Farquhar et al. (2024) extended this idea and introduced discrete semantic entropy to work in black-box settings without access to token probabilities.

Semantic entropy is shown to perform better than standard entropy and outperforms SOTA tools based on model self-evaluation and embedding regression (Kadavath et al., 2022). However, the limitation with this approach is the bidirectional NLI-based semantic clustering. NLI is designed to identify the presence of an entailment or contradictory relationship between two pieces of text. Linguistic phenomena can be complex and nuanced (Naik et al., 2018), and in this case, such a rigid binary classification can sometimes fail to accurately capture semantic similarity due to its continuous nature. Semantic clustering requires multi-dimensional comparison between text pairs to detect any degree of semantic similarity, regardless of whether they are fully an entailment or a contradiction of one another. Furthermore, NLI has been shown to use lexical properties of the text as the main factor in identifying entailment, while heavily relying on specific words in its classification (Arakelyan et al., 2024). Another major limitation of NLI models is their scalability. These models depend on large-scale transformer-based architectures, making them computationally expensive at inference time (Percha et al., 2021).

Therefore, we introduce an optimized semantic clustering approach for efficient and accurate capturing of potentially complex semantic relationships within generations of an LLM, resulting in an improved hallucination detection performance.

## 3 METHODOLOGY

In this section, we provide a detailed description of our approach to automatic black-box hallucination detection in LLMs. To determine semantic equivalence, we apply a fully automated non-prompt based clustering approach, followed by the black-box version of the entropy calculation (Farquhar et al., 2024) to determine the level of uncertainty in the outputs of the LLM. An illustration of the methodology is shown in Figure 1.

**Notation and Problem Statement** The main task involves automatic detection of hallucination in NLG, particularly for QA benchmarks. The process involves prompting an LLM with a question, denoted as $q$, with a generation, $g$, representing the output. To leverage the idea of uncertainty within generations in LLMs, the LLM is prompted $P$ times, resulting in $G = \{g_1, g_2, \ldots, g_P\}$. We concatenate $q$ with each $g_i \in G$, with a separator token, $[SEP]$, between them, to create a representative string '$q \circ [SEP] \circ g_i$', represented by $s_i, \forall g_i \in G$. To detect potential hallucination, we

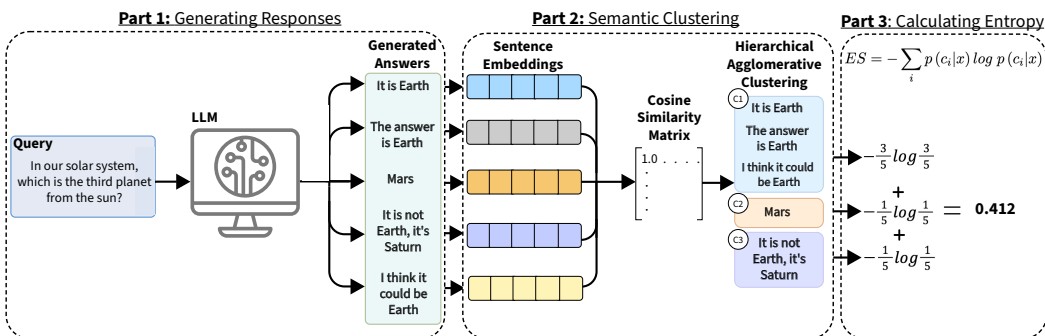

Figure 1: Illustration of our proposed Natural Language Generation hallucination detection framework, involving our optimized semantic clustering approach of multiple generations to calculate semantic entropy. **Part 1** involves generating multiple generations to the same question. **Part 2** then processes the generations and clusters them using sentence embeddings and hierarchical agglomerative clustering. **Part 3** calculates the overall entropy score using the generated clusters.

generate a sentence embedding, using a sentence similarity model, $Emb$, which results in $Emb(s_i)$ with a dimension of $d$.

**Iterative Generation of Outputs** The first part involves iteratively prompting the LLM $P$ times with the question $q$, resulting in multiple generations of the same query. These generations are independent of each other, ensuring that subsequent LM responses are not related to previously generated responses.

**Generating Embeddings** To generate an embedding for every generated answer, we first concatenate $q$ with every $g_i$ with a separator token between them, resulting in $s_i$, to ensure that each $g_i$ is captured within the context of $q$. Text embedding $Emb(s_i)$ are generated by a transformer-based model fine-tuned on the sentence similarity task. Cosine similarity (Rahutomo et al., 2012) is used to estimate the extent of similarity between embeddings as shown below:

$$\text{cos\_sim}(Emb(s_i), Emb(s_j)) = \frac{\langle Emb(s_i), Emb(s_j)\rangle}{\|Emb(s_i)\| \cdot \|Emb(s_j)\|} \tag{1}$$

In this case, cosine similarity is a suitable measure for capturing the overlap between two semantic embeddings due to:

- Focus on direction: It primarily focuses on direction rather than magnitude by emphasizing the angle, $\theta$, between two vectors to calculate similarity (Mikolov et al., 2013).

- Applicability to high-dimensional vectors: Due to the high dimensionality of embeddings, sparsity becomes somewhat of an issue, but with the focus being mainly on $\theta$, cosine similarity is able to capture semantic similarity regardless of the dimensionality (Turney and Pantel, 2010).

- Length-invariant normalization: Normalization disregards any potential differences in lengths, effectively capturing the semantic relationship between the two vectors (Turney and Pantel, 2010).

**Hierarchical Agglomerative Clustering** We employ hierarchical agglomerative clustering to partition the responses into an optimal number of groups. Initially, each embedding $\{Emb(s_1), Emb(s_2), \ldots, Emb(s_P)\}$ forms its own cluster, denoted as $C_1, C_2, \ldots, C_P$, where $C_i = \{Emb(s_i)\}$. The algorithm proceeds iteratively, merging the closest clusters based on a distance function, $dis(C_i, C_j)$, which is defined according to a chosen linkage criterion. The distance threshold, in this case, is set to 0.05 throughout the paper. This process continues until a predefined stopping condition is met. Single linkage may inadvertently connect unrelated clusters, whereas complete linkage is overly sensitive to outliers (Ramos Emmendorfer and de Paula Canuto, 2021).

To mitigate these issues, we adopt average linkage, offering a more balanced distance measure. The distance between embeddings $s_i$ and $s_j$ is defined as:

$$dis(Emb(s_i), Emb(s_j)) = 1 - \text{cos\_sim}(Emb(s_i), Emb(s_j))$$

The pseudocode of the algorithm is provided in Appendix B.

**Agglomerative Clustering Creates More Uniform Partitions** We show that, compared to bidirectional NLI-clustering, hierarchical agglomerative clustering can generate more homogeneous clusters. For instance, consider the example: $q$ = 'In our solar system, which is the third planet from the sun?' and $G$ = ['It is Earth', 'The answer is Saturn','The answer is Earth', 'Mars', 'It is not Earth, it's Saturn', 'I think it could be Earth']. Ideally, we should obtain three clusters representing {Earth, Saturn, Mars}. Clusters obtained from agglomerative clustering are shown in Fig. 2 (a). NLI clustering output is illustrated in Fig. 2 (b). Evidently, agglomerative clustering correctly partition the answers into 3 clusters, whereas NLI results in 5 individual clusters.

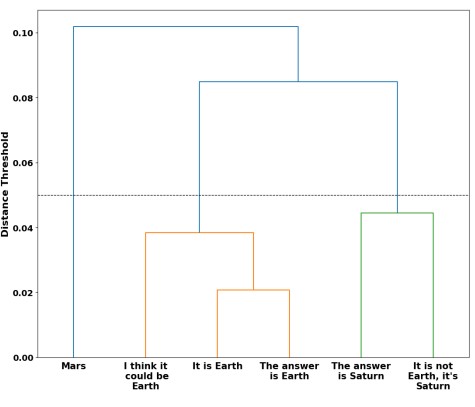

(a) Dendogram of Our Clustering.          (b) Bidirectional NLI Clustering.

Figure 2: Visualization of clusters obtained through agglomerative and NLI based clustering for the same sample.

Our approach successfully identifies that 'I think it could be Earth' belongs with 'It is Earth','The answer is Earth', and 'It is not Earth, it's Saturn' belongs with 'The answer is Saturn', while Bidirectional NLI failed to do so. If we examine the second case, we see that 'The answer is Saturn' is a straightforward affirmative statement, while 'It is not Earth, it's Saturn' consists of two parts: one negating Earth as the answer and the other confirming Saturn as correct. Therefore, in the bidirectional entailment comparison, 'It is not Earth, it's Saturn' entails 'The answer is Saturn', since it logically implies Saturn as the answer, but the reverse is not true because the negation of Earth is not mentioned in the latter statement. In this case, our focus is to cluster based on the final intended answer, without being influenced by other elements of the response, and bidirectional NLI clustering fails to accomplish this.

**Complexity Analysis** Our approach consists of three steps, a) generating sentence embeddings, b) calculating the similarity between embeddings, and c) clustering the generated embeddings. For the first step, we consider that each input question has $P$ answers, which involves tokenization and a forward pass through transformer model. This step has cost $O(P \cdot L^2 \cdot d)$, where $L$ is the number of tokens in the answer, and $d$ is the dimensionality of the resulting embedding. Computing the pairwise cosine similarity between the embeddings cost $P(P-1)/2$ comparisons, and taking into consideration the dimensionality of the embeddings, this amounts to a complexity of $O(P^2 \cdot d)$. Finally, agglomerative clustering has a complexity of $O(P^2 \cdot log\ P)$. The overall complexity of our framework is assessed by adding the cost of individual steps $O(P \cdot L^2 \cdot d) + O(P^2(d + log\ P))$.

**Scalability Analysis** To compare the scalability of our clustering approach with the NLI-based clustering approach, we perform a scalability analysis by reporting the runtime of both approaches over a varying number of generations. For this analysis, we recreate the NLI-based approach using

the DeBERTa-large model [1], as detailed by Kuhn et al. (2023). The results show that our approach is significantly better than NLI-based clustering.

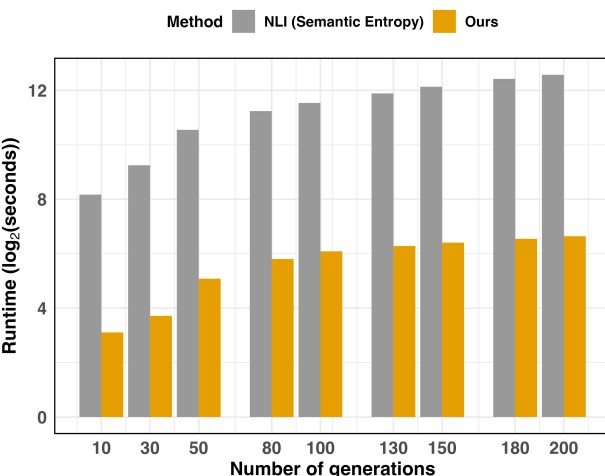

Figure 3: Runtime Analysis of NLI and agglomerative clustering over varying number of generations.

**Calculating Entropy Score** Entropy score of semantic clusters is calculated as shown in Eq. 2.

$$ES = -\sum_i p\left(c_i|x\right) log\ p\left(c_i|x\right) \tag{2}$$

Our formulation is designed for black-box hallucination detection, i.e., we do not need access to internal model state(s) or token probabilities. Hence, entropy can be calculated by using only output tokens.

## 4 EXPERIMENTS

We demonstrate the effectiveness of our approach through a comprehensive experimental set-up.

**Data** The proposed approach is evaluated using four widely-used QA datasets from the literature. These include TriviaQA (Joshi et al., 2017), a trivia-style QA dataset, and Natural Questions (NQ) (Kwiatkowski et al., 2019), which consists of questions derived from Google searches; both are closed-book datasets typically featuring short, one or two-word answers. Additionally, SQuAD (Rajpurkar et al., 2016), a general knowledge open-book QA dataset with longer answers, and BioASQ (Tsatsaronis et al., 2015), a life sciences QA dataset containing either binary (yes/no) or long sentence answers, are utilized. Representative samples for each dataset are provided in Appendix A.

**Models** The proposed methodology is applied to several SOTA LMs, including Llama 2 (Touvron et al., 2023), Mistral (Jiang et al., 2023), and Falcon (Almazrouei et al., 2023). Specifically, the focus is on fine-tuned and instruction-tuned versions, such as Llama-2-7b-chat, LLaMa-2-13b-chat, Falcon-7b-instruct, and Mistral-7b-instruct. To show that the approach works with any *off-the-shelf* LM, no additional fine-tuning is done; instead, the open-source pretrained versions and their corresponding tokenizers available on the Hugging Face website are utilized.

**Comparison with Robust Baselines and SOTA** The proposed approach is compared against four methods as implemented by Farquhar et al. (2024)[2]. In addition to the current SOTA **semantic entropy**, a comparison is made with a supervised **embedding regression** approach (Kadavath et al.,

---

[1]https://huggingface.co/microsoft/deberta-large-mnli
[2]https://github.com/jlko/semantic_uncertainty

2022), which uses a regression model trained on LLM hidden states to predict hallucinations. For baselines, the approach is compared to **naive entropy**, which calculates entropy without accounting for semantic similarity across answers that may use different words or phrases to describe the same concept. Additionally, a comparison is made with **p(true)** (Kadavath et al., 2022), which employs a few-shot prompt-based method to estimate the accuracy of LM outputs.

**Automated Ground-Truth Label** A single "best answer" for each question is generated by setting the model temperature to 0.1. To automatically assess the correctness of LLM-generated output against the ground truth, a semantic similarity measure is used, following the automatic clustering approach proposed in this paper, which incorporates both semantic and cosine similarity for comparison. Embeddings for the ground truth and model answers are generated using the *all-MiniLM-L6-v2* model, chosen for its effectiveness in capturing semantic similarity, particularly in the main experimental clustering setup described in Section 3. The generated response is classified as accurate if the cosine similarity between the embeddings exceeds 0.95, while lower values indicate hallucination.

**Evaluation Metric** In line with prior work, the Area Under the Receiver Operating Characteristic Curve (AUROC) is used as the primary evaluation metric. The ROC curve plots the true positive rate against the false positive rate across various thresholds, making AUROC an appropriate measure for this binary classification task. An AUROC score approaching 1 indicates a strong relationship between the entropy measure and hallucination, whereas an AUROC of 0.5 suggests no meaningful relationship. Higher AUROC values signify better performance.

## 5 RESULTS

Results (Table 1 ) indicate that the proposed approach consistently outperforms baselines in nearly all model-dataset combinations. Specifically, compared to the SOTA semantic entropy approach, the proposed method achieves improvements of up to 7.6% on TriviaQA, 9.3% on NQ, 9.1% on SQuAD, and 4.8% on BioASQ.

For datasets like TriviaQA, NQ, and SQuAD, which feature short responses, the approach excels in capturing subtle semantic differences in minimal inputs. The use of advanced sentence embeddings allows for a deeper understanding of semantic nuances, enhancing clustering performance even in concise textual contexts. The results demonstrate the effectiveness of the proposed method in identifying semantic relationships between generated answers, producing an entropy score that serves as an informative indicator of potential hallucination.

It is important to note that the results for the BioASQ dataset are relatively higher for both the proposed approach and the semantic entropy approach compared to other datasets. This can be attributed to the fact that some answers are binary (yes/no) (Appendix A.4). Such binary responses intuitively simplify the separation and clustering process, unlike other datasets where variations in wording can lead to more complex semantic distinctions.

### 5.1 ABLATION STUDIES

An empirical analysis is conducted to determine the optimal values for various hyperparameters, algorithms, and transformer models used in the experiments.

**Number of Generations** Number of generations ($P$) is an important factor to consider to achieve optimal results. To observe the impact of $P$ on AUROC, we experimented with $P$ values in the range $\{2, 4, 6, 8, 10, 12, 14\}$ across the four datasets. Fig. 4a shows that AUROC values generally increase with an increase in $P$. However, when $P > 10$, the increase is limited and the AUROC starts to level off. Consequently, we set $P = 10$ through our experiments. Apart from achieving the best AUROC, a lower $P$ also reduces the inference costs associated with a higher number of generations.

**Cosine Similarity Threshold for Clustering** We experimented with similarity thresholds in the range $\{0.70, 0.80, 0.85, 0.90, 0.95\}$. The experimental results are shown in Fig. 4b. The results indicate that higher similarity thresholds improve clustering effectiveness, leading to higher AUROC scores across all datasets. Therefore, we use the threshold of 0.95 in our experiments. Choosing a threshold past 0.95 decreases performance, as it imposes a threshold that is too rigid, negatively

Table 1: Evaluation of hallucination detection on open-form QA datasets and 4 representative LLM models. AUROC values are reported. Best performance for each experiment is highlighted in bold.

| Models | Methods | Datasets | | | |
| --- | --- | --- | --- | --- | --- |
| | | TriviaQA | NQ | SQuAD | BioASQ |
| Llama-2-7b-chat | p(True) | 0.642 | 0.646 | 0.607 | 0.786 |
| | Embedding Regression | 0.631 | 0.578 | 0.621 | 0.714 |
| | Naive Entropy | 0.731 | 0.723 | 0.715 | 0.680 |
| | Semantic Entropy | 0.763 | 0.739 | 0.764 | 0.870 |
| | **Ours** | **0.807** | **0.832** | **0.830** | **0.928** |
| LLaMa-2-13b-chat | p(True) | 0.788 | 0.731 | 0.711 | 0.773 |
| | Embedding Regression | 0.695 | 0.698 | 0.592 | 0.732 |
| | Naive Entropy | 0.701 | 0.695 | 0.655 | 0.603 |
| | Semantic Entropy | 0.803 | 0.742 | 0.754 | 0.881 |
| | **Ours** | **0.810** | **0.759** | **0.845** | **0.915** |
| falcon-7b-instruct | p(True) | 0.630 | 0.518 | 0.535 | 0.403 |
| | Embedding Regression | 0.733 | 0.656 | 0.633 | 0.842 |
| | Naive Entropy | 0.767 | 0.732 | 0.649 | 0.697 |
| | Semantic Entropy | 0.786 | 0.736 | 0.710 | 0.861 |
| | **Ours** | **0.807** | **0.821** | **0.797** | **0.909** |
| mistral-7b-instruct | p(True) | 0.758 | 0.730 | 0.643 | 0.757 |
| | Embedding Regression | 0.681 | 0.598 | 0.615 | 0.797 |
| | Naive Entropy | 0.764 | 0.739 | 0.687 | 0.765 |
| | Semantic Entropy | 0.793 | **0.788** | 0.733 | 0.882 |
| | **Ours** | **0.869** | 0.785 | **0.771** | **0.925** |

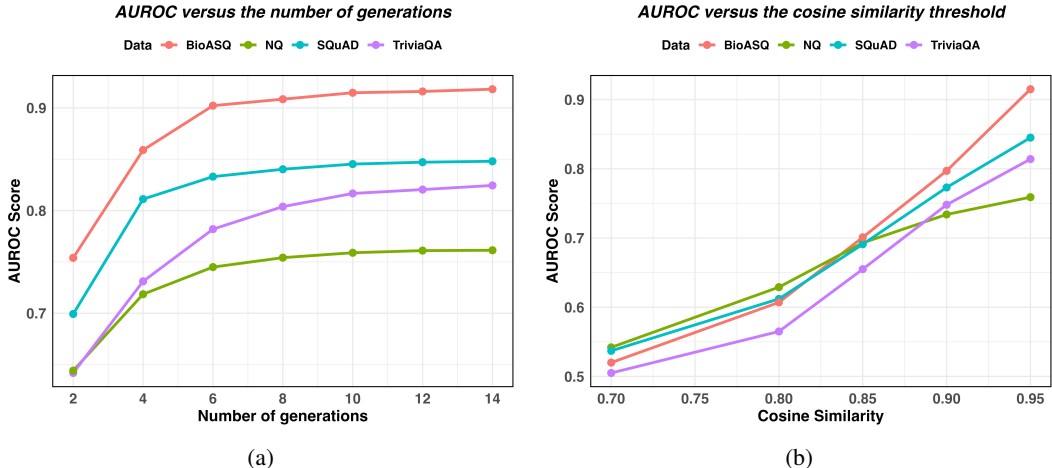

(a)                 (b)

Figure 4: Ablation experiments of LLaMa-2-13b-chat on all datasets for (a) Different number of initial generations. (b) Sensitivity of cosine similarity threshold used for semantic clustering.

impacting the quality of the resulting clusters. This is further illustrated on the TriviaQA dataset in Appendix D.

**Sentence Transformer Model for Semantic Similarity Clustering** To effectively capture semantic similarity between clusters, there are several models that produce meaningful semantically rich embeddings for comparison. To test their effectiveness for our set-up, we experimented with the most popular models (based on download statistics) fine-tuned for the sentence similarity task found

on Hugging Face, including {*all-MiniLM-L6-v2*[3], *all-mpnet-base-v2*[4], *Alibaba-NLP/gte-large-en-v1.5*[5], *paraphrase-multilingual-MiniLM-L12-v2*[6]}. We report the results on LLaMa-2-13b-chat and TriviaQA dataset in Fig. 5. Fig. 5a present the AUROC scores achieved across different models. Additionally, we also show the model efficiency by comparing their runtime in Fig. 5b. Results demonstrate that *all-MiniLM-L6-v2* performed the best in accuracy and runtime efficiency.

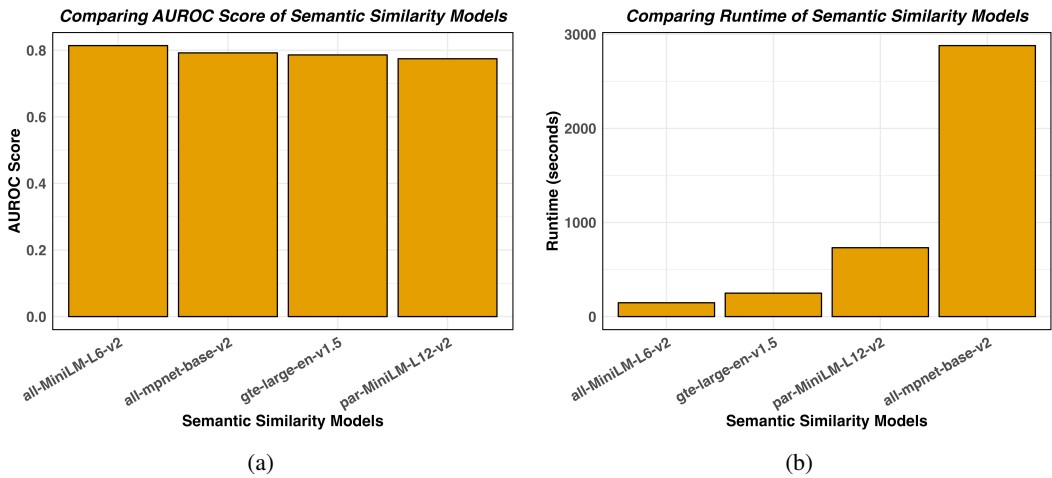

(a)                                         (b)

Figure 5: (a) AUROC results when using different sentence similarity models. (b) Runtime analysis for generating embeddings using each model.

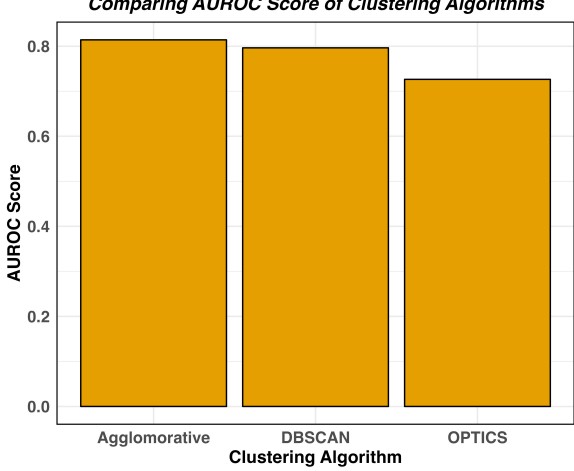

Figure 6: Comparison of AUROC obtained with clustering algorithms.

**Clustering Algorithm** To determine the optimal clustering algorithm based on the cosine similarity comparison between the embeddings, we experimented with Density-Based Spatial Clustering of Applications with Noise (DBSCAN), and Ordering Points to Identify the Clustering Structure (OPTICS), to compare their performance with that of the Agglomerative Hierarchical clustering. As shown in Fig. 6, when experimenting with the LLaMa-2-13b-chat and TriviaQA dataset, we achieved AUROC scores of 0.796, 0.726, and 0.814, respectively. In this case, clustering achieves optimal performance, while detection performance shows a slight decline with the use of other clustering algorithms.

---

[3]https://huggingface.co/sentence-transformers/all-MiniLM-L6-v2

[4]https://huggingface.co/sentence-transformers/all-mpnet-base-v2

[5]https://huggingface.co/Alibaba-NLP/gte-large-en-v1.5

[6]https://huggingface.co/sentence-transformers/paraphrase-multilingual-MiniLM-L12-v2

## 6 CONCLUSION

Hallucination detection is an essential topic to effectively understand and evaluate the reliability and accuracy of LLMs. Automating this process and adapting it to proprietary black-box models is important, particularly due to their increasing integration and prevalence in many contexts. Such explorations play a major role in enhancing the overall trustworthiness of such models. This work proposes an enhanced entropy-based black-box hallucination detection framework by applying an efficient and scalable semantic clustering approach using sentence embeddings and hierarchical agglomerative clustering. We apply this approach to several types of QA datasets, and demonstrate that this approach is effective on free-form NLG data in comparison with state-of-the-art baselines. In the future, we hope that this exploration can be extended to other NLG tasks, to understand its efficiency and applicability at detecting hallucination in different contexts.

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

# A    SAMPLES FROM QA DATASETS

## A.1    TRIVIAQA

**Question:** What was the name of the Oscar-winning song performed by Audrey Hepburn in 'Breakfast at Tiffany's'?
**Answer:** Moon River

**Question:** Late English criminal Bruce Reynolds masterminded which infamous robbery, which he later referred to as his 'Sistine Chapel ceiling'?
**Answer:** Great Train Robbery

## A.2    NQ

**Question:** Who is the actress that plays Aurora in Maleficent?
**Answer:** Elle Fanning

**Question:** Who did Rome fight against in the Punic Wars?
**Answer:** Carthage

## A.3    SQUAD

**Context:** The university is the major seat of the Congregation of Holy Cross (albeit not its official headquarters, which are in Rome). Its main seminary, Moreau Seminary, is located on the campus across St. Joseph lake from the Main Building. Old College, the oldest building on campus and located near the shore of St. Mary lake, houses undergraduate seminarians. Retired priests and brothers reside in Fatima House (a former retreat center), Holy Cross House, as well as Columba Hall near the Grotto. The university through the Moreau Seminary has ties to theologian Frederick Buechner. While not Catholic, Buechner has praised writers from Notre Dame and Moreau Seminary created a Buechner Prize for Preaching.

**Question:** Which prize did Frederick Buechner create?
**Answer:** Buechner Prize for Preaching

**Context:** All of Notre Dame's undergraduate students are a part of one of the five undergraduate colleges at the school or are in the First Year of Studies program. The First Year of Studies program was established in 1962 to guide incoming freshmen in their first year at the school before they have declared a major. Each student is given an academic advisor from the program who helps them to choose classes that give them exposure to any major in which they are interested. The program also includes a Learning Resource Center which provides time management, collaborative learning, and subject tutoring. This program has been recognized previously, by U.S. News & World Report, as outstanding.

**Question:** What was created at Notre Dame in 1962 to assist first year students?
**Answer:** The First Year of Studies program

## A.4    BIOASQ

**Question:** What is the Daughterless gene?
**Answer:** The daughterless (da) gene in Drosophila encodes a broadly expressed transcriptional regulator whose specific functions in the control of sex determination and neurogenesis have been extensively examined.

**Question:** Is the FIP virus thought to be a mutated strain for the Feline enteric Coronavirus?
**Answer:** Yes

## B    Clustering Algorithm Psuedocode

---

**Algorithm 1:** Clustering Algorithm with Average Distance

---

**Input:** set of sequences $S = \{s_1, s_2, \ldots, s_P\}$; embedding model $Emb$; distance threshold $thresh$

**Output:** Set of clusters $C$

1  Initialize empty set of clusters $C = \{\}$;

2  **foreach** *sequence* $s_i \in S$ **do**

3  |    Compute embedding $Emb(s_i)$;

4  **foreach** *sequence* $s_i \in S$ **do**

5  |    Initialize a new cluster $c_i = \{s_i\}$;

6  |    **foreach** *cluster* $c \in C$ **do**

7  |    |    Initialize cumulative distance $total\_dis = 0$;

8  |    |    **foreach** *sequence* $s^{(c)} \in c$ **do**

9  |    |    |    Retrieve embedding $\mathbf{Emb}^{(c)} = Emb(s^{(c)})$;

10 |    |    |    Compute cosine similarity:

$$\text{cos\_sim} = \frac{\langle \mathbf{Emb}(s_i), \mathbf{Emb}^{(c)} \rangle}{\|\mathbf{Emb}(s_i)\| \cdot \|\mathbf{Emb}^{(c)}\|}$$

|    |    |    Compute distance: $dis = 1 - \text{cos\_sim}$;

11 |    |    |    Accumulate the distance: $total\_dis \leftarrow total\_dis + dis$;

12 |    |    Compute average distance (average linkage):

$$\text{avg\_dis} = \frac{total\_dis}{|c|}$$

|    |    **if** *avg\_dis* $\leq$ *thresh* **then**

13 |    |    |    Merge $s_i$ into cluster $c$: $c \leftarrow c \cup \{s_i\}$;

14 |    |    |    **break** (from the inner loop);

15 **return** *clusters* $C$;

---

## C    Implementation details

We use Hugging Face to access transformer models and most datasets throughout the experiments. For BioASQ, we use the training dataset from Task B in the 2023 BioASQ challenge[7]. Primary hyper-parameters to consider are: number of generations ($P$), which we set to $P = 10$, generated by setting the model temperature to 1.0, to keep it consistent with the baselines. Additionally, for automatic semantic clustering, we use the *all-MiniLM-L6-v2* model to generate embeddings, and a cosine similarity threshold of 0.95 (distance of 0.05) for clustering.

## D    Higher Cosine Similarity Threshold Reduces AUROC

Figure 7 shows the AUROC score on the TriviaQA dataset. Using a stringent similarity cutoff ($> 0.95$) forces only highly similar embeddings to be clustered together-this reduces the scope for clustering semantically similar sentences which could be differently phrased.

## E    Code Availability

We provide the code for our approach in the supplementary material.

---

[7]http://participants-area.bioasq.org/datasets/

**AUROC versus the cosine similarity threshold - TriviaQA**

Figure 7: Variation in AUROC as a function of cosine similarity cutoff. The plot is generated with LLaMa-2-13b-chat on TriviaQA. The plot demonstrate the sensitivity of cosine similarity threshold used for semantic clustering.