# OpenReview forum: "Scalable and Enhanced Hallucination Detection in LLMs using Semantic Clustering"
_ICLR.cc/2025/Conference — ICLR 2025 Conference Withdrawn Submission_

### Official Review · Reviewer_QWpw · 2024-10-28

**Soundness:** 3
**Presentation:** 3
**Contribution:** 2
**Rating:** 3
**Confidence:** 3

**Summary:**

This paper proposes an enhanced semantic entropy method for detecting factual errors by improving the semantic the clustering method, it utilizes sentence embedding to capture nuanced semantic properties in a high-dimensional context, followed by hierarchical clustering. Experimental results across multiple benchmarks demonstrate that this approach outperforms the existing state-of-the-art methods.

**Strengths:**

1. The presentation is clear, and the method is easy to follow. This paper incorporates semantic similarity among sampled generations' embeddings to enhance existing semantic entropy, which is an improved version of semantic entropy.
2. The performance is good. Results in Figure 1 demonstrate that this method outperforms existing approaches in most cases.

**Weaknesses:**

1. Lack of Comparison with Other Hidden-State-Based Methods: While the paper compares the proposed method with several entropy-based baselines, it would benefit from comparisons with other hallucination detection approaches based on hidden states, such as INSIDE, Haloscope, and similar frameworks [1][2][3].

2. Limitations in Threshold Sensitivity: Although the paper discusses the effects of cosine similarity thresholds, it lacks a detailed sensitivity analysis. The 0.5 coefficient mentioned is only validated on the QA benchmarks evaluated, which may limit its generalizability to tasks in other domains, such as mathematical reasoning tasks (e.g., GSM8K) [4] and factuality tasks (e.g., TruthfulQA) [5].

References:
[1]. INSIDE: LLMs' Internal States Retain the Power of Hallucination Detection. Chen et al., 2024.
[2]. Discovering Latent Knowledge in Language Models Without Supervision. Burns et al., 2023.
[3]. HaloScope: Harnessing Unlabeled LLM Generations for Hallucination Detection. Du et al., 2024.
[4]. Training Verifiers to Solve Math Word Problems. Cobbe et al., 2021.
[5]. TruthfulQA: Measuring How Models Imitate Human Falsehoods. Lin et al., 2021.

**Questions:**

Refer to the weakness

---

### Official Review · Reviewer_Vv68 · 2024-10-28

**Soundness:** 2
**Presentation:** 1
**Contribution:** 1
**Rating:** 3
**Confidence:** 5

**Summary:**

This paper proposes a semantic clustering framework to improve semantic entropy hallucination detection. The semantic clustering framework includes creating clusters of sentence embeddings then using these clusters to improve the accuracy of semantic entropy method in detecting hallucination.

**Strengths:**

- The similarity-based clustering method improves the accuracy of Semantic Entropy in detecting hallucinations in open-domain Question Answering tasks.
- The method is generalisable to four LLMs.

**Weaknesses:**

## Major

- The novelty of the contribution is very limited.
	- The paper proposes using sentence embedding similarity followed by clustering as opposed to an NLI-based method to improve the accuracy of a semantic entropy method in detecting hallucination. The semantic entropy method is not novel, nor is the pairwise sentence similarity matrix approach in NLP.
- The use of `all-MiniLM-L6-v2` for evaluation is not properly supported beyond the ablation study.
	- The authors should provide an analysis to show that `all-MiniLM-L6-v2` is a strong model to calculate semantic similarity measures. For instance, the authors may check the correlation between the similarity score given by this model and human evaluation as opposed to just comparing it with other sentence embedding models.
	- The authors should also justify why the existing metrics (e.g., subspan Exact Match and F1 score) are not sufficient to identify inaccurate answers (or hallucinations), requiring AUROC to be included.
	- The authors report the evaluation using AUROC. However, this depends highly on how well-calibrated the evaluation embedding model is. The authors should provide an analysis to show the calibration of the similarity score (with respect to the correctness of the answer).
- Missing baselines:
	- Lin, Z., Trivedi, S., & Sun, J. (2023). Generating with Confidence: Uncertainty Quantification for Black-box Large Language Models. Trans. Mach. Learn. Res., 2024.
- Lack of discussion on the ablation study results
	- The ablation studies are presented as a compilation of results which require more interpretation.
	- For instance, the comparison of the clustering algorithms does not include any reasoning why one method works better than the others.
- The presentation is very lacking. See the following "minor" and "very minor" weaknesses subsections.

## Minor

- L17-19: There is a logical gap between the uncertainty-based method sentence and the semantic clustering.
- L39-41: This requires a citation (or rephrasing). I believe that all hallucination detection works have always been focusing on the accuracy of the generated response, not the phrasing.
- L190-191: Provide more details about the text embedding model.
	- What is the model size?
	- Was the model pretrained on another corpus?
	- What is the sentence similarity task?
	- What is the size of the sentence similarity dataset?
- L250-251: Provide more explanation into why the bidirectional NLI model failed to do the clustering. Provide more intuition and analysis into this result.
- L270-271: Which "results" are you referring to?
- Equation 2: c_i, x, and p(c_i|x) were not previously defined.
- L333: `all-MiniLM-L6-v2` was introduced very late given that the authors use it not only for evaluation but also for the proposed methodology.
- L432-433: The authors should support the decision to choose these four embedding models.

## Very minor (e.g., typos)

- Inconsistent uses of abbreviations "LLM" and "LM".
- Quite a lot of redundant and less critical explanations (e.g., L122-125, L197-208).
- L81-82: "Proliferation of LMs in real-world scenarios [...] is significantly limited ...". This sentence contradicts the first sentence of the abstract.
- L84: what "approaches"?
- L86: "Black box methods depend on the output text generated by LMs". This does not read very well, I believe the authors meant to say that black box methods only require the output text.
- L94-98: These sentences do not seem to contribute much to the point this paragraph trying to convey.
- Figure 1: Equations inside the Part 3 box are very blurry
- L209-218: The presentation of this paragraph can be improved. The authors can choose to finish discussing about the distance function first (along with the equations), and then move on to discuss the linkage.
- Figure 2b: Typo "It is not **Eath**"
- L319-328: This paragraph is very hard to parse. Consider enumerating the baseline methods with 1), 2), 3), and so on.
- L329-342: These two paragraphs can be combined together as AUROC is calculated from the thresholded cosine similarity.
- Spell out numbers less than 10.

**Questions:**

See the weaknesses section above.

---

### Official Review · Reviewer_iDbG · 2024-11-03

**Soundness:** 3
**Presentation:** 3
**Contribution:** 2
**Rating:** 3
**Confidence:** 4

**Summary:**

This study presents an optimized semantic clustering framework for automated hallucination detection in LLMs, using sentence embeddings and hierarchical clustering. The proposed method can enhance both scalability, speed, and performance compared to existing approaches across different LLM models. The experiements in various QA tasks shows the effectiveness of the method.

**Strengths:**

1. The proposed hallucination detection method is both intriguing and straightforward.
2. The experiments reveal that the method significantly improves scalability, achieving a 60-fold increase in speed compared to previous methods. Additionally, results from various QA datasets indicate an enhancement in the accuracy of hallucination detection.
3. The paper is well-organized, making it easy to understand and follow.

**Weaknesses:**

1. The contribution of the proposed method appears incremental, showing high similarity to the semantic entropy method. Although there is an enhancement in performance, the primary distinction lies in the clustering approach. Unlike the semantic entropy method, which uses an NLI model to cluster answers, the proposed method leverages embeddings from <question, answer> pairs derived from NLI models.
2. The authors do not sufficiently analyze the clustering performance within the proposed framework. For instance, evaluating the clustering effectiveness through human verification could provide additional insights into its accuracy.
3. The evaluation methodology lacks robustness. The authors choose answers generated at a temperature setting of 0.1 as the 'best answer,' without justifying why adopted such as a setting.
4. According to Figure 4(b), there is a positive correlation between the similarity and AUROC. However, a similarity value of 0.95 does not achieve the maximum AUROC value. It is unclear what the upper limit of the maximum AUROC value is.

**Questions:**

See Weakness.

---

### Official Review · Reviewer_z9hR · 2024-11-04

**Soundness:** 3
**Presentation:** 4
**Contribution:** 3
**Rating:** 8
**Confidence:** 4

**Summary:**

The paper proposes a semantic clustering framework to improve uncertainty estimation and hallucination detection in LLMs. Their method is built upon the idea of semantic entropy (Kuhn et al. 2023), which involves clustering the generated responses based on semantic similarity and estimating the overall entropy for the semantic clusters. One limitation of semantic entropy approach is that it uses natural language inference (NLI) model to estimate semantic similarity, which is computationally expensive and hard to capture nuanced semantic properties. To address this issue, the authors propose to extract sentence embeddings for each generated response and then cluster the responses by hierarchical agglomeration clustering. The experiment results on multiple QA datasets show that their method is effective for hallucination detection, measured by AUROC.

**Strengths:**

- Hallucination detection is an important and timely topic. Their work can contribute to the development of more reliable and trustworthy models.
- The paper is well-written, clear and easy to follow. The authors provide a comprehensive and critical analysis of relevant literature, which helps me understand the work in the context very well.
- The proposed approach is intuitive, simple to implement yet effective, computationally efficient and can be widely applicable to various LLMs.
- The empirical results are impressive. The new method outperforms all the baseline hallucination detection approaches across different QA datasets and language models.

**Weaknesses:**

- Notations in equation (2) are not clearly explained.
- The presentation of tables and figures could be improved. The fonts and styles used in the Figure 3/4/5/6 and Table 1 (especially the titles) are not formal enough.
- Lack of quantitative evaluation of the quality of semantic clusters produced by the proposed framework

**Questions:**

- In section 4, the authors mention that a single “best answer” for each question is generated by setting temperature to 0.1. This experiment setup is confusing to me. Why did you run the evaluation in an unsupervised way instead of using the gold answers from the datasets?

---

### Note · Authors · 2024-11-25

I have read and agree with the venue's withdrawal policy on behalf of myself and my co-authors.